# Thinking about My Existence during COVID-19, I Feel Anxiety and Awe—The Mediating Role of Existential Anxiety and Life Satisfaction on the Relationship between PTSD Symptoms and Post-Traumatic Growth

**DOI:** 10.3390/ijerph17197062

**Published:** 2020-09-27

**Authors:** Katarzyna Tomaszek, Agnieszka Muchacka-Cymerman

**Affiliations:** Department of Psychosomatic, Institute of Psychology, Pedagogical University of Cracow, Podchorążych 2, 30-084 Kraków, Poland; katarzyna.tomaszek@up.krakow.pl

**Keywords:** existential anxiety, PTSD stress symptoms, life satisfaction, post-traumatic growth, stress

## Abstract

Background: The global outbreak of COVID-19set new challenges and threats for every human being. In the psychological field it is similar to deep existential crises or a traumatic experience that may lead to the appearance or exacerbation of a serious mental disorder and loss of life meaning and satisfaction. Courtney et al. (2020) discussed deadly pandemic COVID-19 in the light of TMT theory and named it as global contagion of mortality that personally affected every human being. Such unique conditions activate existential fears as people start to be aware of their own mortality. Objective: The main aim of this study was to test the mediating effect of existential anxiety, activated by COVID-19 and life satisfaction (SWLS) on the relationship between PTSD symptoms and post-traumatic growth (PTG). We also examined the moderated mediating effect of severity of trauma symptoms on life satisfaction and existential anxiety and its associations with PTG. Method: We conducted an online survey during the peak of the COVID-19 outbreak in Poland. The participants completed existential anxiety scale (SNE), life satisfaction scale (SWLS), IES-R scale for measuring the level of PTSD symptoms and post-traumatic growth inventory (PTGI). Results: The effect of PTSD on PTG was found to be mediated by existential anxiety and life satisfaction. We also confirmed two indirect effects: (1) the indirect effect of PTSD on PTG via existential anxiety and life satisfaction tested simultaneously; (2) the indirect effect of life satisfaction on PTG through severity of trauma symptoms. An intermediate or high level of PTSD level was related to less PTG when low and full PTSD stress symptoms strengthened PTG experiences. Conclusions: A therapeutic intervention for individuals after traumatic experience should attempt to include fundamental existential questions and meaning of life as well as the severity of PTSD symptoms. The severity of traumatic sensations may affect the relationship between life satisfaction and post-traumatic growth.

## 1. Introduction

### 1.1. The Concept of Existential Anxiety in Psychology and Pandemic COVID-19

Existential anxiety affects people in different life situations. As Temple et al. [1] stated it is inevitable for human beings to experience anxiety that concerns death, isolation, emptiness, freedom, and meaninglessness. As such, it changes individuals’ interactions and relation to self, others, and the world. As Menzies [2] stated, existential anxiety may be invoked by former decisions, acts, choices, and may bring on existential guilt or regret due to the sense of a lost chance. Tillisch [3] hypothesized that existential anxiety is a core human issue, and it contains three domains: (1) anxiety about fate and death related to the fear of absolute annihilation that might result from the threat of death combined with awareness of human mortality; (2) anxiety of emptiness and meaninglessness that refers to individual’s threat of non-being and the worthlessness of life; (3) anxiety about guilt and condemnation defined as the result of the threat to moral and ethical self-affirmations, when the individual does not live up to their own moral and ethical standards. Awareness of the inevitability of death is inherently connected with terror management theory (TMT) by Greenberg et al. [4]. The TMT assumes that in life people strive to achieve a higher self-esteem, because it functions as a kind of mechanism that protects against fear of death [4]. The combination of both—the awareness of inevitable death and the will to survive—arouses in man existential fear and the potential for paralyzing terror [5]. This potential for terror is guided by two buffers: the individual’s outlook and self-esteem. In TMT existential anxiety usually stays at the latent level, however some circumstances activate it. This is especially the case in situations that provide expressiveness of death—i.e., terminal disease.

The research of Stynska et al. [6] draws attention to the fact that fear is the most important and most basic element of human existence. In such a way, man reacts to uncertainty. Recently, Courtney et al. [7] have applied the TMT theory into pandemic situation, and proposed terror management health model (TMHM). The authors concentrate on motivations to people’s engaging in certain behaviors in the context of serious health threats—i.e., during a pandemic. According to TMHT, the COVID-19 pandemic heightens the consciousness of human mortality, activates death-related thoughts, and forces people to make decisions and engage in health behaviors. Additionally, people may reduce perceived death and health vulnerability risk caused by pandemic in two ways by maladaptive or adaptive health behaviors. Maladaptive health behaviors are similar to distinguished in TMT theory two defense mechanisms (1) proximal defenses which function to remove the threat of mortality from consciousness; (2) distal defenses, when thoughts of death are activated, but not consciously accessible, so people may concentrate on cultural values or their ideological beliefs and think that the threat is not personally relevant to them so they do not to follow the authorities’ recommendations to limit the spread of the virus.

### 1.2. The Relationship between Existential Anxiety, Trauma, Life Satisfaction, and PTG

As Nowak-Dziemianowicz [8] wrote, the reason for existential anxiety experienced by a person may be the family environment and his/her behavior patterns learned from his/her youth. The experience of this kind of anxiety often appear as a result of traumatic experience that force people to re-organize their lives and to confront the conflicts inherent in life and towards death [9]. Recently, Emanuel et al. [10] described COVID-19 pandemic as a situation that has created an environment in which existence is more fragile and existential fears or terror rises in people. Such ultimate life threats may also activate the process of making an existential imbalance of life. Difficult events generate disruptive effects, therefore they disrupt the capacity of the value and meaning systems, self-esteem, and social relationships to perform their normal anxiety-buffering functions [11]. Trauma is often explained as an existential injury because it threatens and devastates basic human needs and goals [12,13]. It is worth adding that according to the existential approach, authenticity encourages people to accept the painful aspects of human existence and pushes them forward into the future, diminishing existential anxiety [2]. What is more, overwhelming distress, amplified by prior trauma, may cause inability to cope with death and resort to defensive reactions [10].

Sezer et al. [14] found that sub-dimensions of existential anxiety (meaninglessness anxiety and isolation anxiety) are significant predictors of low life satisfaction and self-esteem. According to the authors, having a meaning and purpose in human life, and not feeling isolated from others are the core variables contributing to life satisfaction for adolescents and those beginning young adulthood. It is also worth noting that Stynska et al. [6] draw attention to the positive accent of anxiety, in which its higher level should be considered a higher possibility of revealing a person’s potential goal. From such a perspective, death awareness is characterized as bittersweet, since it is connected either with negative emotions (anxiety, fear, and experiential avoidance and the idea that we are essentially alone in life), but also with inspiration, creativity, innovation, positive change, and the contribution to something greater than one’s self [15,16]. Callely [17] found that existential anxiety moderates the association between post-traumatic stress symptoms and post-traumatic growth by triggering meaning-making behavior or actions that help make sense of life events. As Yalom [18] stated, the idea of death has the potential to saves us as it may plunge us into more authentic life modes, and it enhances our satisfaction in the living of life. In addition, COVID-19 has been identified as a serious cause of psychological existential crises that affect all areas of human life on all levels. According to Emanuel et al. [10] pandemic COVID-19 has brought harrowing experiences to many human beings, not only for those it has killed and people have many reason to feel fears and existential crises. Specifically, in the psychological fields it increases the level of distress, anxiety, and other mental health problems as well as loneliness, social withdrawal, and lack of professional support [19,20,21]. Bobdey at al. [22] claim that even after the physical symptoms of the disease are over, people may suffer from social and mental problems. As Pargament wrote, “negative life events are often accompanied by fundamental changes in the person’s value system (view of reality) and in his/her experience of the self” [23]. Therefore, it seems that difficult life experiences, for example the COVID-19 pandemic, may activate and maintain existential anxiety or change the pleasure from human life or the mental balance, which may modify our beliefs about the sense of our own existence, but also the significance of past difficult events.

### 1.3. Trauma, Post-Traumatic Growth, and Life Satisfaction

The four core post-traumatic stress symptoms disorder (PTSD) are defined by the American Psychological Association (APA) (2013) as intrusion, negative alterations in mood and cognition, avoidance of trauma-related stimuli, and hyperarousal. PTSD results in extreme distress and functional impairment, and as a long-term consequence it causes poor quality of life and lower level of life satisfaction, serious health problems, e.g., depression, chronic physical conditions (such as back and neck pain, headaches, heart disease, or stroke) [24,25,26]. However, trauma may also be connected with positive consequences and change for the better. The theory of post-traumatic growth (PTG) experienced by the individual is based on a positive evaluation of the past difficult traumatic situations and, as a result, acquiring a new sense of it and change in all areas of life [27]. Tedeschi et al. [28] suggest that people may emerge from trauma or adversity having achieved positive personal growth. Originally the construct of PTG consists of five domains: Appreciation of Life, New Possibilities, Relating to Others, Personal Strength, and Spiritual Change [29]. The authors point out the importance of ruminating on an event by an individual, i.e., a process in which a person has recurrent thoughts about an experienced traumatic event. This traumatic event evaluation may be positive or negative [30]. By analyzing the self-perspective of interpreting the meaning of a trauma, a person gives it a new and positive value and sense. Southwick et al. [31] described PTGI as a stress-induced increase in psychological benefits.

Research shows that an individual’s experience of a traumatic event may be crucial for mental disorders or ineffectiveness in various areas of life at a later stage in every plane, both family and professional [32], because trauma causes other problems and shapes the level of satisfaction an individual derives from life. Zsigmond et al. [33] found a positive relationship between coping strategies, emotional severity of post-traumatic stress symptoms, social support contribution, and post-traumatic growth; however, the association differs in relation to severity in PTSD symptoms. The authors stated that post-traumatic growth has a weak to moderate association with quality of life. Despite the fact that theoretically trauma and post-traumatic growth remain strongly related, the empirical verification of such association still remains unclear. Some authors discovered negative relationships between these two constructs [34] whereas others found positive associations [35] or non-linear association (quadratic relationship) [36,37]. Recently, it was found that the relationship between PTSD symptoms and PTGI may differ in order of the severity of the traumatic experience or the category of traumatic event (bereavement, physical assault, and rape) [38] or time after trauma experience with increased PTG during the first 18 months and stable level after this time [39]. Additionally, some authors suggest that the above-mentioned relationship may be mediated or moderated by other psychological constructs—e.g., the effect of PTSD symptoms on PTG was fully mediated by resilience—and this mediating effect was also moderated by childhood trauma (the more childhood traumatic experiences, the greater the mediating effect of resilience) [26] or socio-demographic characteristics—e.g., women reporting more posttraumatic growth than men as the mean age increases [40].

## 2. Purpose of the Study

This study had three main goals: (1) to test the mediating effect of existential anxiety and life satisfaction on the relationship between PTSD stress and post-traumatic growth (PTG); and (2) to examine the moderated mediating effect of severity of trauma symptoms on life satisfaction and existential anxiety and its associations with PTG.

We hypothesized that existential anxiety and life satisfaction would mediate the association between PTSD and PTG, and that these mediators may interact together on this relationship. Furthermore, we hypothesized that the severity of the PTSD stress would moderate the association between existential anxiety and PTG, and between life satisfaction and PTG.

Despite the fact, that the connectedness between existential anxiety, PTSD symptoms, post-traumatic growth, and life satisfaction have been already tested by several researchers there are several uniqueness of our study. First, the unique time of data collection—e.g., the COVID-19 pandemic. The COVID-19 pandemic is a serious cause of distress that may modulate the relationship between abovementioned psychological variables. To our knowledge, none of the prior research have been done during a pandemic, and only a few tested the abovementioned variables all together. What is more, we examined the moderated mediating effect of severity of trauma symptoms on life satisfaction and existential anxiety and its associations with PTGI, that has not been yet tested. Finally, it seems very important, that we tested the associations between past trauma symptoms and actual stress of individuals connected with high death thought accessibility, as COVID-19 is a serious threat to human health and life.

## 3. Materials and Methods

### 3.1. Study Population

Data were collected with an online survey in Poland during the first peak of the outbreak of COVID-19 pandemic (March and April 2020), from 199 university students (155 females, 29 males), aged from 18 to 48 years (M = 21.92 years, SD = 4.70). The participants did not have psychiatric diagnosis of PTSD. They completed the IES-R scale in order to test the severity of PTSD symptoms during the COVID-19 pandemic. We categorized all samples into four groups following the Creamer et al. [41] proposition, with a cutoff of 1.5 (equivalent to an IES-R total score of 33). 16.8% of the sample had low level of PTSD symptoms, and 9.3% scored average level of it, so they had partial PTSD or some of the symptoms. 8.4% of the participants scored high level of these symptoms, that indicate the probable diagnosis of PTSD. Half of the respondents had full PTSD symptoms, that indicate serious problems with mental health and problems with proper immune system functioning (*n* = 110, 51.4%) (see Table 1). The traumatic experiences reported by students concerned various kinds of categories, most often the loss of somebody close (*n* = 70). Most often participants experienced trauma from 1 to 5 years ago (*n* = 82, 41.2%) (see Table 2).

### 3.2. Instruments

**Impact Event Scale- Revised Scale (IES-R)** by Weiss and Marmar [42]. The tool consists of 22 statements and is intended to assess the subjective stress caused by a traumatic event. It consists of three PTSD dimensions: intrusion—when a person experiences recurrent trauma images; agitation—in which a person has high anxiety and difficulty concentrating; and avoiding—when the individual has to put a lot of effort into getting rid of the thoughts or emotions that accompany trauma. According to the instructions, a person first describes a traumatic event in life and then, using a five-point Likert scale (0–4), evaluates the symptoms he or she experiences. In the current study, the internal consistency of the measure was equal to Cronbach’s α = 0.92.

**Existential Anxiety and Fear Scale—short version (SNE)** by Juros [43]. The tool measures the construct of existential anxiety. The SNE consists of 25 items according to which a person answers on a seven-point Likert scale. It allows to test the total score of SNE (an overall level of existential anxiety) and its three dimensions: (1) fear of guilt and meaninglessness; (2) fear of emptiness and condemnation; (3) fear of fate and death. In our study, we used the overall level of existential anxiety. Cronbach’s α for total score was equal to 0.94.

**Satisfaction with life scale (SWLS)** by Diener et al. [44]. It measures specific life satisfaction domains and global cognitive judgments. The tool consists of five statements where a person answers on a seven-point Likert scale (1—strongly disagree, 7—strongly agree). The higher the score achieved by the individual, the higher the life satisfaction. In the current study, Cronbach’s α was equal to 0.85.

**Post-traumatic growth inventory (PTGI)** by Tedeschi and Calhoun [28]. It is a 21-item scale with a six-point Likert scale (0—strongly disagree, 5—strongly agree). The higher the score, the higher the intensity of the positive changes. The inventory measures four factors that contribute to post-traumatic development: changes in self-perception, changes in relationships with others, greater appreciation of life, and spiritual changes. Cronbach’s α for PTGI total score was equal to 0.92.

### 3.3. Data Analysis

The descriptive statistics were calculated to summarise the demographic characteristics of the respondents. Pearson’s correlation analysis were estimated in order to present the associations between all the study variables. Estimation of mediating effects was tested by using PROCESS for SPSS Macro [45]. In order to investigate the indirect-mediation effects and moderation mediation effects, bootstrap method was applied with 5000 resamples, and bias-corrected 95% confidence intervals (CI).

### 3.4. Ethical Consideration

This study was approved by the Commission of the Ethics Committee of the Pedagogical University in Cracow (WP BS-642/P/2019/20).

## 4. Results

The descriptive statistics for each of the four groups divided by severity of PTSD symptoms are presented in Table 3. According to Kruskal–Wallis Test, significantly higher PTSD symptoms and existential anxiety was found in the group with full PTSD symptoms (group 4). The level of life satisfaction was similar in all groups. The comparisons of PTG results indicated that group 3 had significantly lower post-traumatic growth than group 4 (see Table 3).

The results of Pearson’s analysis revealed the significant and positive associations between the PTSD symptoms (IES-R) and existential anxiety indicators (Pearson’s correlation coefficient r ranged between 0.21 to 0.36), while life satisfaction negatively correlated with IES-R score (*r* = −0.15; *p* = 0.039), and PTG was not connected with traumatic symptoms. Life satisfaction was positively correlated with PTG (Pearson’s *r* = 0.23, *p* = 0.002), and PTG was also significantly and negatively connected only with one existential anxiety indicator e.g., anxiety of fate and death (*r* = −0.37, *p* < 0.0001) (see Table 4).

To examine the mediation and moderation effects, we applied macro PROCESS by Hayes (see Figure 1). In model 1, multiple mediation effect was tested with two mediators (model 6 in the Hayes PROCESS); in model 3, moderation mediation model was examined (model 15 in the Hayes PROCESS). Table 5 presents the estimated regression coefficients of the direct effects for the examined models and the mediation-moderation effects. We tested one sub-dimension of existential anxiety—Anxiety of fate and death (SNE3)—as a potential mediator, because only this variable significantly correlated with PTSD symptoms (see Table 4).

According to the results for model 1, individuals with more PTSD symptoms expressed more existential anxiety (β = 0.39; *p* < 0.0001; CI95% = 0.41 to 0.87—path a1). Surprisingly, PTSD symptoms were not connected to life satisfaction (β = −0.04, SE = 0.03; CI95% = −0.07 to 0.04—path a2). However, existential anxiety as a mediator, significantly affected both life satisfaction (β = −0.34; *p* < 0.0001; CI95% = −0.10 to −0.04), and PTG (β = 0.20; *p* = 0.018; CI95% = 0.03 to 0.26—path b1). The indirect effect of PTSD symptoms on PTG via existential anxiety was significant (β = 0.09, SE = 0.04, CI95% = 0.01 to 0.18), and via life satisfaction was insignificant (β = −0.01, SE = 0.02, CI95% = −0.06 to 0.02). Furthermore, the indirect effect of two mediators tested simultaneously (existential anxiety and life satisfaction) was significant (β = −0.04, SE = 0.03, CI95% = −0.08 to −0.01) (see Table 6, Figure 1).

In model 2 we confirmed the simple mediation effect of existential anxiety on the relationship between life satisfaction and PTG (β = −0.07, SE = 0.03, CI95% = −0.15 to −0.02). The results of moderation-mediation analysis tested in model 3 revealed that the interactive effect of existential anxiety and severity of traumatic symptoms on PTG was insignificant (β = −0.02, CI 95% = −0.12 to 0.08; *p* = 0.765). However, the indirect effect of life satisfaction on PTG through severity of trauma symptoms was significant trauma (β = 0.84; CI95% = 0.31 to 1.37; *p* = 0.765) (see Table 5, Figure 2). The test revealed that severity of trauma symptoms may impact the experience of life satisfaction and post-traumatic growth. Specifically, the relation between life satisfaction and positive growth in the group with full PTSD symptoms (group 4) is strong and positive, and in low (group 1) and high (group 3) PTSD the symptoms are strong and negative.

## 5. Discussion

The presented study aimed to test the mediation effects of existential anxiety (SNE), activated by the COVID-19 pandemic and life satisfaction (SWLS), on the post-traumatic stress (PTSD) and post-traumatic growth (PTG) relationship. Secondly, we also examined the mediation effect of existential anxiety and the moderated mediating effect of severity of traumatic symptoms on the association between SWLS and PTG.

Despite the fact that the relationship between PTSD, SWLS, and PTG had already been tested in several studies, to our knowledge the investigation that includes the aforementioned variables combined with severity of trauma and existential anxiety has not yet been conducted. Additionally, such an investigation during the COVID-19 pandemic has created special conditions for analyzing the relationship between these variables because all the participants experienced a real threat to their health and life as it was the time of population-wide crisis [10]. Additionally, many recently conducted studies have confirmed that COVID-19 pandemic is connected with diminished well-being and escalated chronic symptoms of psychological distress and other mental disorders [46,47].

The results indicated that the effect of PTSD on PTG is mediated by existential anxiety and life satisfaction, and that these psychological characteristics simultaneously impacted the relationship between the independent (PTSD) and dependent (PTG) variables. Additionally, the relationship between life satisfaction and post-traumatic growth is moderated by the severity of traumatic symptoms. Our also study confirmed previous findings which indicated that the association between PTSD and PTG may be a curvilinear relationship (a U association) [26,36,37]. However, in our study the U association was not inverse. Tsai et al. [37] found an inverted U-shaped relationship between PTSD symptoms and PTG among veterans. A meta-analysis conducted by Shakespeare–Finch and Lurie–Beck [36] revealed a significant linear relationship between PTG and PTSD symptoms, but also a curvilinear relationship that depends on trauma type and age. In our studies U-shaped relationship meant that an intermediate or high level of PTSD level seemed to lead to less PTG, when low and full PTSD stress symptoms strengthened PTG experiences. Some of the past results suggested that trauma reaction, because of its complexity, may be a vehicle for personal growth after a difficult event. For example, in Jin et al.’s [48] study, higher levels of trauma and post-traumatic distress among earthquake survivors were associated with greater post-traumatic growth. The authors explained this relationship in two ways (1) PTG occurs when the trauma has been upsetting enough for person to promote engagement in a positive outlook about the event, but not too overwhelming for survivors to handle; (2) PTG and PTSD have a shared psychological ‘engine’ that sets them in motion, and PTG reflect a cognitive adaptation process among those who experience post-traumatic stress disorders in response to their disaster (a positive reinterpretation). Our findings may be also the results of factors that were not controlled in our study; however, they may modify the relationship between PTSD and PTG—e.g., personality characteristics and maladaptive coping strategies [49], resilience [26], and unsupportive social network [50], time passing after traumatic event [51].

Similarly to prior studies conducted by Schubert et al. [35], we revealed that individuals with more severe post-traumatic symptoms show more PTG than those with low PTSD characteristics. Furthermore, we also found that the severity of trauma experiences may influence the association between life satisfaction and PTG level. In individuals with low PTSD symptoms in the abovementioned relationship, it was negative, and in the group with full PTSD symptoms it was positive. These findings may shed light on the mechanism that underlies the trauma symptoms and its meaning to the individuals. Perhaps the key here is the currently felt life satisfaction resulting from the acceptance of past life experiences of the individual and the current existential fear. Zhang et al. [47] revealed that the severity of COVID-19 and restriction connected with this pandemic negatively impacted life satisfaction and this effect depended on the chronic medical issues of the individuals In our studies traumatic stress symptoms may be consider as chronic mental disease, and such symptoms strengthened existential fear connected to COVID-19, diminished life satisfaction, and via these mediators reduced PTG. Additionally, Dymecka et al. [52] found that the relationship between stress felt during pandemic COVID-19 and life satisfaction was mediated by sense of coherence, and the fear of COVID-19 acted as a buffer by weakening the relationship between stress and a sense of coherence. Looking at the results, it is worth noting that the respondents faced new difficult and threatening events during the study, which were probably significant for their satisfaction with life and interpretations of previous difficult events. The significant correlations between PTSD symptoms and existential fear felt during COVID-19 pandemic may suggest that people after trauma must also cope with existential crises that include interventions in the psycho-spiritual aspects of the human psyche, such as fear of fate and death; anxiety of guilt and meaninglessness and fear of emptiness and condemnation. Understanding the interface of the psychological and spiritual dimensions may contribute to a more efficient therapeutic practice. Spirituality relates to quality of life and mental well-being.

The findings should also be interpreted in the light of recently developed Terror Management Health Model for Pandemics by Courtney et al. [7]. Similarly to TMT theory, from the perspective of TMHM model people engage in different two defensive tactics to cope with the fundamental threat of their mortality and health vulnerability e.g., (1) proximal defenses which are defined as threat-focused attempts to push death/health vulnerability outside of consciousness—denial or avoidance mechanism; and (2) distal defenses which are connected with response to non-conscious death thoughts that are not accessible and derived automatically [7,53]. Notwithstanding, it is possible that PTSD symptoms in some way may protect against confrontation with current life-threats and existential crises due to the fact that focusing on past and known difficult events may be easier than dealing with loneliness caused by forced isolation, the possibility of infection, and the death or loss of somebody close because of the SARS-CoV-2 virus. In light of the abovementioned terror management health model, such actual reactions on past trauma during new deadly pandemic may be recognized as a form of proximal defenses—e.g., (1) reducing perceived vulnerability to health or death threat by for example reinforcing the self- belief that a person has already experienced so many difficulties, therefore, is strong enough to deal with COVID-19; (2) denial perceived vulnerability to health or death threat by strong concentration on PTSD symptoms caused by past trauma and ignoring the personal risk of infection of the SARS-CoV-2 virus. Such an interpretation seems to be also in accordance with Li et al.’s [54] statement that a lack of knowledge about COVID-19 and the treatment methods negatively impact on the mental health issue, as it causes fear, anxiety, and insecurity regarding the future and the meaning of one’s own life. Kar et al. [21] have provided some guidelines for people with long-term illness during COVID-19 to maintain their mental health. The authors concentrated on avoiding a distressing situation, and an exposure to media, as well as maintaining an online relationship with friends and family, and trying to think positively. The findings of our studies suggest that struggling with past trauma distress symptoms is connected with existential crises indicated by high existential anxiety felt during pandemic COVID-19, and both may diminish life satisfaction and PTG. In order to minimize coronavirus anxiety, Kecmanovic [55] advises practicing tolerating uncertainty, accepting anxiety as an integral part of human experience, and trying to transcend it by connecting it to one’s own life’s purpose and sources of meaning, sharing negative feelings with others (especially in spirituality relations), trying to increase self-awareness by getting adequate sleep, exercising regularly, and employing relaxation techniques. In accordance with our findings, a proper strategy to cope with COVID-19 distress is also the dynamic-dialectical process of spirituality as described by Yang et al. [23]. It consists of reflection on the experience of being part of a greater whole and being blessed with a positive sense of life, as well as increasing awareness and accepting the total loss of meaning and trying to (re)discover it in a wholly new form. A person needs to prepare for the moments of discontinuity and be able to discuss with others their own existential dilemma and anxiety.

## 6. Study Limitation

Our findings should be interpreted along with taking into account several study limitations. The most important limitation seems to be the fact that the respondents chose one of the traumas on the list, but the loss of a loved one can have different meanings—for example the loss of a parent or grandfather. In our study, we also did not control important sociodemographic characteristics that are important in experiencing PTG—e.g., gender, age, higher level of education, ethnicity, the severity of traumatic experiences [37,38,48] or social support [54]. In addition, the time at which the trauma occurred may significantly affect the test results; some respondents indicated that the traumatic event occurred about 3 months ago, whereas others 5 years ago. Another limitation is that mainly women participated in our study; in the group of women, trauma may have completely different overtones—in connection with cultural, biological, and developmental factors—than in the group of men, so it is important that in later studies the groups are equally numerous.

## 7. Conclusions

Despite its limitations, the findings of this study increase the understanding of the association between post-traumatic symptoms and growth as well as between PTGI and other psychological constructs that are important aspects of functioning in a distress-related situation, i.e., the COVID-19 pandemic. The most important conclusions from the research are that: (1) existential anxiety may modulate the relationship between PTSD symptoms and PTG; (2) PTSD symptoms may impact indirectly PTG via existential anxiety and life satisfaction; (3) the severity of traumatic sensations can affect the relationship between life satisfaction and post-traumatic growth. A therapeutic intervention for individuals with existential trauma that may be caused by many contributors (e.g., the COVID-19 pandemic) should attempt to include fundamental existential questions and meaning of life as well as the severity of PTSD symptoms. An individual affected by existential anxiety about the emergence of traumatic situations that may cause PTSD has more difficulty reaching any strategies for coping with stress in a constructive way, using mature defense mechanisms [56,57].

## Figures and Tables

**Figure 1 ijerph-17-07062-f001:**
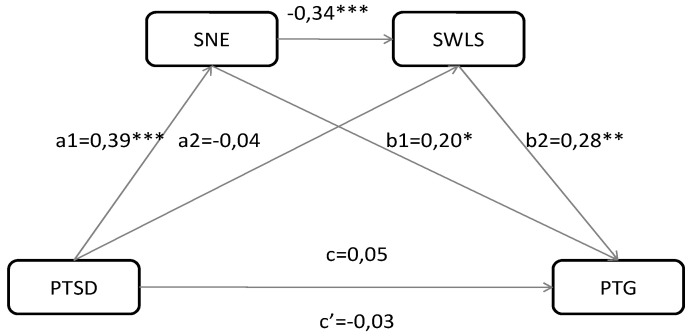
Mediation effect model among symptoms of trauma (PTSD), existential anxiety (SNE), life satisfaction (SWLS), and post-traumatic growth (PTG). * *p* < 0.05; ** *p* < 0.001; *** *p* < 0.0001.

**Figure 2 ijerph-17-07062-f002:**
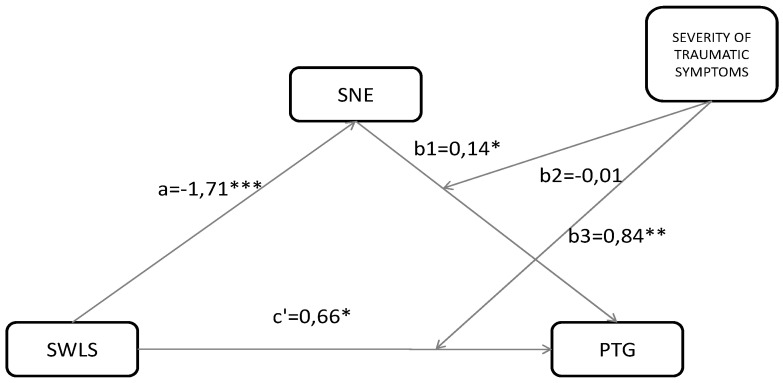
Mediation-moderation model among severity of PTSD symptoms, existential anxiety (SNE), life satisfaction (SWLS), and post-traumatic growth (PTG). * *p* < 0.05; ** *p* < 0.001; *** *p* < 0.0001.

**Table 1 ijerph-17-07062-t001:** Prevalence of traumatic symptoms.

IES-R Score	Number (%)	Group
1–23	36 (16.8%)	Group 1. Low traumatic symptoms
24–33	20 (9.3%)	Group 2. Average traumatic symptoms
25–37	18 (8.4%)	Group 3. High traumatic symptoms
38 and above	110 (51.4%)	Group 4. Full PTSD symptoms

**Table 2 ijerph-17-07062-t002:** Demographic characteristics of the participants (*n* = 199).

Variables	Group 1.	Group 2.	Group 3.	Group 4.	Total Sample
**Age (M,SD)**	21.69 (4.20)	21.00 (2.10)	21.67 (4.42)	22.05 (5.06)	21.92 (4.70)
**Gender**					
Female	27	17	17	94	155 (84.9)
Male	9	3	1	16	29 (15.1)
Lack of data					15 (7.5)
**Type of traumatic experience**					
Loss of somebody close	7	13	8	42	70 (35.2)
Work and financial problems	5	0	5	30	40 (20)
Family problems or divorce	7	1	4	13	25 (12.6)
Sickness or disability	12	3	0	17	32 (16.1)
Violent event (assault or accident)	2	3	1	8	14 (7.1)
Other	3	0	0	0	3 (1.5)
Lack of data					15 (7.5)
**Time since exposure to trauma**					
From last month to one year	5	4	4	30	43 (21.6)
1–5 years	14	13	10	45	82 (41.2)
>5years	17	3	4	35	59 (29.6)
Lack of data					15 (7.6)

**Table 3 ijerph-17-07062-t003:** Descriptive statistics (M,SD) for the study variables.

Variables	Group 1		Group2		Group 3		Group 4		Kruskal–Wallis Test	*p*	Post Hoc ^a^
	M	SD	M	SD	M	SD	M	SD			
1. IES-R	16.69	6.79	28.25	2.57	35.17	1.15	52.46	11.27	142.31	<0.0001	1–2,3,4; 2–3,4; 3–4
2. Anxiety of guilt and meaninglessness	35.86	13.42	42.75	6.12	27.11	9.30	45.42	12.91	32.27	<0.0001	1–2,3,4; 2–3; 3–4
3. Anxiety of emptiness and condemnation	27.97	8.85	29.40	6.16	25.17	4.85	34.62	10.39	25.11	<0.0001	1,2,3–4
4. Anxiety of fate and death	24.06	8.76	28.95	7.37	15.89	7.14	27.51	9.99	24.97	<0.0001	1,2–3; 3–4
5. Existential anxiety total score	87.89	26.28	101.10	10.33	68.17	19.03	107.51	26.52	34.84	<0.0001	1–2,3; 2–4;3–4
6. Life satisfaction	24.79	4.67	24.05	4.75	23.50	6.11	22.64	6.31	4.78	0.189	-
7. PTG	61.36	19.46	65.15	22.07	46.67	21.45	65.41	16.34	13.39	0.004	3–4

Note: ^a^—Games—Howell Test; IES-R—PTSD symptoms, PTG—post-traumatic growth level.

**Table 4 ijerph-17-07062-t004:** Pearson’s correlation matrices.

Variables	1	2	3	4	5	6	7
1. IES-R	-						
2. Anxiety of guilt and meaninglessness (SNE1)	0.36 ***	-					
3. Anxiety of emptiness and condemnation (SNE2)	0.31 ***	0.72 ***	-				
4. Anxiety of fate and death (SNE 3)	0.21 **	0.53 ***	0.22 **	-			
5. Existential anxiety total score (SNE)	0.36 ***	0.94 ***	0.80 ***	0.70 ***	-		
6. Life satisfaction	−0.15 *	−0.36 **	−0.57 ***	0.11	−0.35 ***	-	
7. PTG	0.07	0.02	−0.07	0.37 ***	0.12	0.23 **	-

Note: IES-R—PTSD symptoms, PTG—post-traumatic growth level; * *p* < 0.05; ** *p* < 0.001; *** *p* < 0.0001.

**Table 5 ijerph-17-07062-t005:** Direct effects and moderated-mediation outcomes.

Number of Model	Path	Standardized Coefficients	Unstandardized Coefficients				95%CI	
		Coeff(β)	Coeff(B)	SE	*t*	*p*	LLCI	ULCI
**Model 1**	PTSD → PTG(c)	0.05	0.061	0.09	0.69	0.490	−0.11	0.23
	PTSD → SNE(a1)	0.39	0.641	0.12	4.45	<0.0001	0.41	0.87
	PTSD → SWLS(a2)	−0.04	−0.013	0.03	−0.49	0.626	−0.07	0.04
	SNE→PTG (b1)	0.20	0.141	0.06	2.40	0.018	0.03	0.26
	SWLS → PTG(b2)	0.28	0.928	0.26	3.54	0.001	0.41	1.45
	SNE → SWLS	−0.34	−0.07	0.02	−4.37	<0.0001	−0.10	−0.04
	PTSD → PTG(c’)	−0.03	0.026	0.09	0.28	0.781	−0.16	0.21
**Model 2**	SWLS → PTG(c)		0.66	0.27	2.48	0.014	0.14	1.19
	SNE → PTG(b1)		0.14	0.06	2.59	0.010	0.03	0.25
	Severity of PTSD symptoms → PTG		−0.10	1.26	−0.08	0.936	−2.59	2.39
	SNE * Severity of PTSD symptoms → PTG(b2)		−0.02	0.05	−0.30	0.765	−0.12	0.08
	SWLS * Severity of PTSD symptoms → PTG(b3)		0.84	0.27	3.13	0.002	0.31	1.37

Note: PTSD—symptoms of trauma, SNE—existential anxiety, SWLS—life satisfaction, PTG—post-traumatic growth. * *p* <0.05.

**Table 6 ijerph-17-07062-t006:** Significance test of mediation effects by bootstrapping.

Number of Model	Path	Standardized Indirect Effects	SE	95%LLCI	95%ULCI
Model 1	PTSD → SNE → PTG	0.08	0.04	0.01	0.18
	PTSD → SWLS → PTG	−0.01	0.02	−0.06	0.02
	PTSD → SNE → SWLS → PTG	−0.04	0.02	−0.08	−0.01
Model 2	SWLS → SNE → PTG	−0.07	0.03	−0.15	−0.02

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
