# Peer review of "Thinking about My Existence during COVID-19, I Feel Anxiety and Awe—The Mediating Role of Existential Anxiety and Life Satisfaction on the Relationship between PTSD Symptoms and Post-Traumatic Growth"

_ijerph, 2020, doi:10.3390/ijerph17197062_

Round 1

Reviewer 1 Report

The research is interesting and deals with a current and relevant topic. It is well presented with sufficient literature review and a solid methodology. However, since most of the conclusions are not new but strengthen previous research (as the authors themselves say), the merit of this paper would be much higher if the authors would give some theoretical analysis and explanations to their findings. 

My detailed comments for the authors are:

  1. Please elaborate what is the uniqueness of your research.
  2. Please add a short theoretical discussion on your findings: Explain their meaning on the backdrop of previous research.
  3. Please elaborate on the following issues: 
  • What do your findings add to the existing knowledge, besides strengthening previous research?
  • Please elaborate on your conclusion that "The most important conclusion from your study is that PTSD symptoms may constitute a kind of defense mechanism in realizing the presence of a threat in the current epidemiological situation".

Please tighten the connection between your conclusions to your main aim as you defined it: "The main aim of this study was to test the mediating effect of existential anxiety, activated by COVID-19, on the relationship between life satisfaction, PTSD symptoms and post-
traumatic growth (PTGI)."
In this respect, please tighten the connection between the TMT theory, and your findings.

Author Response

Response: Thank you for your valuable comments we added the suggested elements.

The research is interesting and deals with a current and relevant topic. It is well presented with sufficient literature review and a solid methodology. However, since most of the conclusions are not new but strengthen previous research (as the authors themselves say), the merit of this paper would be much higher if the authors would give some theoretical analysis and explanations to their findings. 

My detailed comments for the authors are:

  1. Please elaborate what is the uniqueness of your research.
  2. Please add a short theoretical discussion on your findings: Explain their meaning on the backdrop of previous research.
  3. Please elaborate on the following issues: 
  • What do your findings add to the existing knowledge, besides strengthening previous research?
  • Please elaborate on your conclusion that "The most important conclusion from your study is that PTSD symptoms may constitute a kind of defense mechanism in realizing the presence of a threat in the current epidemiological situation".

Please tighten the connection between your conclusions to your main aim as you defined it: "The main aim of this study was to test the mediating effect of existential anxiety, activated by COVID-19, on the relationship between life satisfaction, PTSD symptoms and post-
traumatic growth (PTGI)."
 In this respect, please tighten the connection between the TMT theory, and your findings.

Reviewer 2 Report

General comments:

This paper strives to discuss the ways in which existential anxiety and life satisfaction impact PTSD symptoms and post traumatic growth, in the context of COVID-19 by way of administering surveys to university students that assess their levels of existential anxiety, life satisfaction, trauma symptoms and post-traumatic growth. Examining the associations between life satisfaction and existential anxiety scores with post traumatic growth scores is worthy of study. However, there is a discrepancy in the aims of the abstract and the methods; I have detailed these concerns below. The authors would benefit from streamlining the data to show the analyses that are meaningful to study and describing why they are analyzing those data points specifically and what meaningful contribution they have found for the field.

There is an effort in this study to further elucidate psychological factors associated with post-traumatic stress and post-traumatic growth, and there is a perceived importance by the authors that studying this during COVID-19 is significant, however this significance needs further conceptualization as right now the paper seems more to be an evaluation of PTSD and modifying factors that happens to be done during COVID-19. There could be merit in comparing such survey results to future scores or to comparing results to case-controls pre-COVID. This paper would benefit from a radical restructuring.

Title:

  • The title is long and does not succinctly describe the aims of the study, but rather discusses one particular relationship that was assessed. Please consider a title that more broadly summarizes the paper’s main goal.

Abstract:

  • The abstract was largely difficulty to follow and the connection between existential anxiety and it being activated by COVID-19 needs to be further discussed.
  • Line 17 – what is the I in PGTI?
  • Line 17 – 19 The various connections discussed in the abstract are difficult to follow, i.e. the existential anxiety and life satisfaction as well as both above-mentioned variables effects’ of mediation of the effect of PTSD on PTGI. Please elucidate what this means.
  • Line 19 – the authors use the phrase, “above-mentioned variables.” Please clarify
  • Line 19-20 – please clarify the difference between PTSD level and PTSD symptoms.

Introduction:

  • The introduction would benefit from more specific definition of existential anxiety as soon as it is introduced. The introduction is very wordy and difficult to follow. Focus on the variables you are studying and their interconnectedness. There is much superfluous background.
  • The introduction is difficult to follow. Would recommend reevaluating what you want the reader to understand as it relates to your study and its variables. Currently the introduction is at times too broad and meandering.
  • Line 35 – what is latter referring to?
  • Line 39 – please rephrase this sentence for meaning.
  • Line 43 – what is “it” referring to?
  • Line 52 – please discuss the equation of existential anxiety as a special kind of fear
  • Line 54 – what do you mean by update?
  • Line 58 – please elaborate on what is meant by “higher possibility of revealing”
  • Line 60 – what are these groups referring to?
  • Line 64 – what is “situation of choice”, please describe this
  • Line 68 – this is wordy, please edit for meaning
  • Line 73 – please define meaninglessness anxiety and isolation anxiety
  • Line 77-78 – what do you mean by “positive accent of anxiety” and its “higher level”?
  • Line 86 – discuss the rational for identifying COVID-19 as a serious cause of distress and how this may modulate or effect results of surveys and how to account for the effect of COVID-19 on such results, i.e. is there any research done on prior pandemics and traumatic stress of individuals?
  • Line 95 – what is the “balance of it” referring to it? Please edit
  • 106 – what does “whole of life” mean?
  • Line 108 – what is the “construct” you are referring to?
  • Line 118 – “positive relationship” between what exactly?
  • Further elaborate on the purpose of the study, see bullets below.
  • Line 131 – please discuss what this mediating effect means? Or how you hypothesize this mediation is occurring and what is the importance of understanding it?
  • Line 134 – please describe the connection between life satisfaction on PTGI via existential anxiety; this does not make sense as written.

Methods:

  • Line 144 – When exactly was data collected and why at that time?
  • Line 169 – what is the general level of existential anxiety mean? Is there a non-general level?
  • Line 192 – Please elucidate difference between “higher PTSD symptoms” and “full PTSD symptoms”, what is the difference between “higher” and “full”? are these different scales? Please elaborate.

Discussion:

  • Line 265 - 266 – this is important to highlight, you should highlight this earlier in the study as a focal point of the paper as this seems meaningful. The other analyses seem superfluous and make this paper confusing to fully appreciate.
  • Line 267-268 – how can you fully assess impact of COVID-19 on the data? Is it possible? What is this special condition exactly mean?
  • Line 272 – please clearly define what the independent and dependent variables of your study are
  • Line 276 -277– could you provide context for this curvilinear relationship? Is there a hypothesis for why that is? In your study, is there a hypothesis for why the U association was not inverse and what that means?
  • Line 281 – why do you think the severity influences this association? Please explain what this means and how this information is meaningful to the field
  • Line 286 – what does balance of life experiences mean? Is this in reference to traumatic vs non traumatic experiences?
  • Line 288 – how might facing these events have changed the results? Is it fair to draw a conclusion that this was significant to their satisfaction with life and interpretation of prior trauma?
  • Line 289 – please explain this confirmation as it does not seem warranted
  • Line 290 – please discuss psycho-spiritual more explicitly, and what that means in your study’s context
  • Line 292 – 295 – is it possible to make the leap that PTSD symptoms protect patients from the current stressor of COVID? This seems like an unfounded possibility from your study’s findings
  • Line 299 -312 – this paragraph does not appear helpful to the study, appears to be providing advice for distress, but not related to study’s research goals or analysis
  • Line 318 – the post traumatic growth may be related to the timing of the event, has this been controlled for in some way? or can this be addressed somehow in your study?
  • Line 320 – this is a blanket statement, please discuss what these overtones are
  • Line 323 – 325 – the connection between functioning during a “distress-related situation” seems a big leap, please further elaborate, there may be many other confounding factors
  • Please restructure your conclusion to properly reflect the results you found, as it appears you have made large leaps here, saying PTSD symptoms are a defense mechanism during COVID-19. How does the data support this? Where is this coming from? How is this defense being assessed considering we do not have pre-COVID data?
  • How does a therapeutic intervention of this sort inform an individual’s PTSD treatment?
  • Line 330 – the relation between existential trauma being caused by COVID-19 is a big assumption, as this trauma may have many other contributors.

Figures and tables:

  • The figures are difficult to interpret, please consider restructuring them to better see the arrows. Please discuss what mediation means in the context of your study variables. Provide more context to the figure legends as the actual purpose of studying these mediating effects does not appear clear.
  • Table 5 – there is a slew of data here, I would recommend streamlining this to the variables you highlighted earlier in the introduction. The rest can be made into a supplemental table. Please discuss why you are looking at the various interactions in the way that you have outlined, what is the hypothesized importance of looking at these relationships?
  • Figure 3 is unclear. Is this looking at the moderation of severity of symptoms on life satisfaction and post traumatic growth separately? If so, please separate into two figures.

Author Response

Response: Thank you for your valuable and detailed comments. We agree with many of them and we made many changes fallowing your suggestions. However, with some of them we disagree. A detailed information we have placed under your suggestions.  

We agree with most

General comments:

This paper strives to discuss the ways in which existential anxiety and life satisfaction impact PTSD symptoms and post traumatic growth, in the context of COVID-19 by way of administering surveys to university students that assess their levels of existential anxiety, life satisfaction, trauma symptoms and post-traumatic growth. Examining the associations between life satisfaction and existential anxiety scores with post traumatic growth scores is worthy of study. However, there is a discrepancy in the aims of the abstract and the methods; I have detailed these concerns below. The authors would benefit from streamlining the data to show the analyses that are meaningful to study and describing why they are analyzing those data points specifically and what meaningful contribution they have found for the field.

There is an effort in this study to further elucidate psychological factors associated with post-traumatic stress and post-traumatic growth, and there is a perceived importance by the authors that studying this during COVID-19 is significant, however this significance needs further conceptualization as right now the paper seems more to be an evaluation of PTSD and modifying factors that happens to be done during COVID-19. There could be merit in comparing such survey results to future scores or to comparing results to case-controls pre-COVID. Thispaperwould benefit from a radicalrestructuring.

Title:

  • The title is long and does not succinctly describe the aims of the study, but rather discusses one particular relationship that was assessed. Please consider a title that more broadly summarizes the paper’s main goal.

We disagree with this comment. The title lists the variables that were tested. Any deletion would have to entail a change in structure and dependent.

Abstract:

  • The abstract was largely difficulty to follow and the connection between existential anxiety and it being activated by COVID-19 needs to be further discussed. Thank you it has been changed
  •  
  • Line 17 – what is the I in PGTI? Thank you it has been changed
  •  
  • Line 17 – 19 The various connections discussed in the abstract are difficult to follow, i.e. the existential anxiety and life satisfaction as well as both above-mentioned variables effects’ of mediation of the effect of PTSD on PTGI. Please elucidate what this means. Thank you it has been changed
  • Line 19 – the authors use the phrase, “above-mentioned variables.” Please clarify. Thank you it has been changed
  • Line 19-20 – please clarify the difference between PTSD level and PTSD symptoms.

We used commonly measure for PTSD symptoms level. We added sentence: the level of PTSD symptoms, as the scale allow to measure overall level of PTSD symptoms.

Introduction:

  • The introduction would benefit from more specific definition of existential anxiety as soon as it is introduced. The introduction is very wordy and difficult to follow. Focus on the variables you are studying and their interconnectedness. There is much superfluous background.
  • The introduction is difficult to follow. Would recommend reevaluating what you want the reader to understand as it relates to your study and its variables. Currently the introduction is at times too broad and meandering.
  • Line 35 – what is latter referring to? Thank you it has been changed
  • Line 39 – please rephrase this sentence for meaning. Thank you it has been changed
  • Line 43 – what is “it” referring to? Thank you it has been changed
  • Line 52 – please discuss the equation of existential anxiety as a special kind of fear. Thank you it has been changed
  • Line 54 – what do you mean by update? Thank you it has been changed
  • Line 58 – please elaborate on what is meant by “higher possibility of revealing” Thank you it was our mistake we have changed it
  • Line 60 – what are these groups referring to?  Thank you it was deleted
  • Line 64 – what is “situation of choice”, please describe this. it was deleted
  • Line 68 – this is wordy, please edit for meaning. Thank you it has been changed
  • Line 73 – please define meaninglessness anxiety and isolation anxiety

We cited the results of Sezer et al. studies so we belief that it is unnecessary to defined these terms, it will make the article too long, what is more elements of the definitions of these terms are included in the next sentence

  • Line 77-78 – what do you mean by “positive accent of anxiety” and its “higher level”?

It was described in the next sentence of the article: From such a perspective, death awareness is characterised as bittersweet, since it is connected either with negative emotions (anxiety, fear and experiential avoidance and the idea that we are essentially alone in life), but also with inspiration, creativity, innovation, positive change and the contribution to somethinggreater than one’s self

  • Line 86 – discuss the rational for identifying COVID-19 as a serious cause of distress and how this may modulate or effect results of surveys and how to account for the effect of COVID-19 on such results, i.e. is there any research done on prior pandemics and traumatic stress of individuals?

In this line we did not include any information about it, however it was done by citing other authors in the next few lines (from 88)

  • Line 95 – what is the “balance of it” referring to it? Please edit
  • The word mental balance has been added
  • 106 – what does “whole of life” mean?

We changed it for “change in all areas of life”

  • Line 108 – what is the “construct” you are referring to?

We added PTG

  • Line 118 – “positive relationship” between what exactly?

We disagree that is not clear. The sentence included such information, the expression means that between all listed psychological variables

Zsigmond et al. [34] found a positive relationship between coping strategies, emotional severity of post-traumatic stress symptoms, social support contribution and post-traumatic growth, however the association differs in relation to severity in PTSD symptoms.

  • Further elaborate on the purpose of the study, see bullets below.
  • Line 131 – please discuss what this mediating effect means? Or how you hypothesize this mediation is occurring and what is the importance of understanding it?

We added these information

  • Line 134 – please describe the connection between life satisfaction on PTGI via existential anxiety; this does not make sense as written.

We deleted this mistake

Methods:

  • Line 144 – When exactly was data collected and why at that time? Thank you, it was added
  • Line 169 – what is the general level of existential anxiety mean? Is there a non-general level?

We have changed it for an overall level of existential anxiety

  • Line 192 – Please elucidate difference between “higher PTSD symptoms” and “full PTSD symptoms”, what is the difference between “higher” and “full”? are these different scales?

We addend suchinformation

Discussion:

  • Line 265 - 266 – this is important to highlight, you should highlight this earlier in the study as a focal point of the paper as this seems meaningful. The other analyses seem superfluous and make this paper confusing to fully appreciate.
  • Line 267-268 – how can you fully assess impact of COVID-19 on the data? Is it possible? What is this special condition exactly mean?

Thank you, we added information

  • Line 272 – please clearly define what the independent and dependent variables of your study are

Thank you, it was added

  • Line 276 -277– could you provide context for this curvilinear relationship? Is there a hypothesis for why that is? In your study, is there a hypothesis for why the U association was not inverse and what that means?

We added more detailed information about U shape association between PTSD and PTG, however w havediscussed out non inverse U results in next sentences by describing changes in this relationship connected with severity of the traumatic symptoms and life satisfaction

However in our study the U association was not inverse. An intermediate or high level of PTSD level seemed to lead to less PTG, when low and full PTSD stress symptoms strengthened PTG experiences. Similarly to prior studies conducted by Schubert et al. [36] we revealed that individuals with more severe post-traumatic symptoms show more PTG than those with low PTSD characteristics.

  • Line 281 – why do you think the severity influences this association? Please explain what this means and how this information is meaningful to the field

It has been already explained in the article, so we disagree with this comment:

Furthermore, we also found that the severity of trauma experiences may influence the association between life satisfaction and PTG level. In individuals with low PTSD symptoms in the above-mentioned relationship, it was negative, and in the group with full PTSD symptoms it was positive

  • Line 286 – what does balance of life experiences mean? Is this in reference to traumatic vs non traumatic experiences?

Thank you, we have changed that

  • Line 288 – how might facing these events have changed the results? Is it fair to draw a conclusion that this was significant to their satisfaction with life and interpretation of prior trauma?

Thank you, we added examples of studies to prove our interpretation

  • Line 289 – please explain this confirmation as it does not seem warranted

We add explanation by describing proximal and distal defence from  TMHM model for pandemic

  • Line 290 – please discuss psycho-spiritual more explicitly, and what that means in your study’s context

Understanding the interface of the psychological and spiritual dimensions may contribute to a more efficient therapeutic practice.Spirituality relates to quality of life and mental well-being

  • Line 299 -312 – this paragraph does not appear helpful to the study, appears to be providing advice for distress, but not related to study’s research goals or analysis

We disagree with this comment, we added the statement. In our study we examined population of students that struggle with past trauma distress symptoms and existential anxiety, that is connected with COVID- 19 pandemic. We believe that information about techniques of coping with COVID 19 means as one of existential crisis is reasonable if we consider tested sample and variables.

  • Line 318 – the post traumatic growth may be related to the timing of the event, has this been controlled for in some way? or can this be addressed somehow in your study?

It is a part of study limitation, and we have already included information that time after trauma is important and that we did not control it in this studies, we added in the introduction such information

  • Line 320 – this is a blanket statement, please discuss what these overtones are

We have included variables that may have an impact on the result.

  • Line 323 – 325 – the connection between functioning during a “distress-related situation” seems a big leap, please further elaborate, there may be many other confounding factors
  • Please restructure your conclusion to properly reflect the results you found, as it appears you have made large leaps here, saying PTSD symptoms are a defense mechanism during COVID-19. How does the data support this? Where is this coming from? How is this defense being assessed considering we do not have pre-COVID data?

Thank you, we have changed that

  • How does a therapeutic intervention of this sort inform an individual’s PTSD treatment?
  • Line 330 – the relation between existential trauma being caused by COVID-19 is a big assumption, as this trauma may have many other contributors.

Thank you, we changed this expression

Figures and tables:

  • The figures are difficult to interpret, please consider restructuring them to better see the arrows. Please discuss what mediation means in the context of your study variables. Provide more context to the figure legends as the actual purpose of studying these mediating effects does not appear clear.

We add once more figures in order to better see all their elements

  • Table 5 – there is a slew of data here, I would recommend streamlining this to the variables you highlighted earlier in the introduction. The rest can be made into a supplemental table. Please discuss why you are looking at the various interactions in the way that you have outlined, what is the hypothesized importance of looking at these relationships?

We excluded one model from the article in order to simplify results, and concentrate only on main variables

  • Figure 3 is unclear. Is this looking at the moderation of severity of symptoms on life satisfaction and post traumatic growth separately? Ifso, pleaseseparateintotwofigures.

We decided to exclude this figure from the text

Round 2

Reviewer 2 Report

The authors have addressed my suggestions.